# Citizens, Scientists, and Enablers: A Tripartite Model for Citizen Science Projects

**Rhian A. Salmon** [1,2,3,*], **Samuel Rammell** [3], **Myfanwy T. Emeny** [4] **and Stephen Hartley** [3]

1   Centre for Science in Society, Te Herenga Waka, Victoria University of Wellington, Wellington 6140, New Zealand
2   Te Pūnaha Matatini Centre of Research Excellence, University of Auckland, Auckland 1142, New Zealand
3   Centre for Biodiversity and Restoration Ecology, School of Biological Sciences, Te Herenga Waka, Victoria University of Wellington, Wellington 6140, New Zealand; rammelsamu@myvuw.ac.nz (S.R.); Stephen.Hartley@vuw.ac.nz (S.H.)
4   Wellington City Council, Wellington 6140, New Zealand; Myfanwy.Emeny@wcc.govt.nz
*   Correspondence: Rhian.Salmon@vuw.ac.nz

**Abstract:** In this paper, we focus on different roles in citizen science projects, and their respective relationships. We propose a tripartite model that recognises not only citizens and scientists, but also an important third role, which we call the 'enabler'. In doing so, we acknowledge that additional expertise and skillsets are often present in citizen science projects, but are frequently overlooked in associated literature. We interrogate this model by applying it to three case studies and explore how the success and sustainability of a citizen science project requires all roles to be acknowledged and interacting appropriately. In this era of 'wicked problems', the nature of science and science communication has become more complex. In order to address critical emerging issues, a greater number of stakeholders are engaging in multi-party partnerships and research is becoming increasingly interdisciplinary. Within this context, explicitly acknowledging the role and motivations of everyone involved can provide a framework for enhanced project transparency, delivery, evaluation and impact. By adapting our understanding of citizen science to better recognise the complexity of the organisational systems within which they operate, we propose an opportunity to strengthen the collaborative delivery of both valuable scientific research and public engagement.

**Keywords:** advocacy; biodiversity; conservation biology; citizen science; Great Kererū Count; i-Naturalist; Lion Guardians; participatory science; public engagement with science; wicked problems

## 1. Introduction

A number of models have been proposed for interrogating the design of citizen science. In its infancy, citizen science was perceived to operate primarily between citizens and scientists. At its most basic, the nature of the relationship between these can be 'top-down', where scientists engage with citizens to collect data, or 'bottom-up', where citizens initiate data gathering and research to answer questions of concern [1]. Expanding on this, Bonney, et al. [2] describe a continuum of citizen science from (i) contributory projects, designed by scientists in which members of the public are primarily contributing data; to (ii) collaborative projects, also led by scientists, but in which members of the public have a role in research design, adaptation or dissemination; to (iii) co-created projects, which are designed and implemented by both scientists and members of the public together. Adding to each end of the spectrum, Shirk, et al. [3] further recognise contractual projects, where public participants "contract" research from scientists, and collegial contributions, where "non-credentialed individuals conduct research independently with varying degrees of expected recognition by institutionalised science and/or professionals". However, all of these models, focus primarily on the relationship between two parties: citizens and scientists (Figure 1A). From hereon, we refer to these as "roles", acknowledging that an

individual person involved in a given project may fill more than one role, and conversely, a given role may be occupied by a diversity of people (see also Voronov and Weber [4] to explain our reference to people rather than actors).

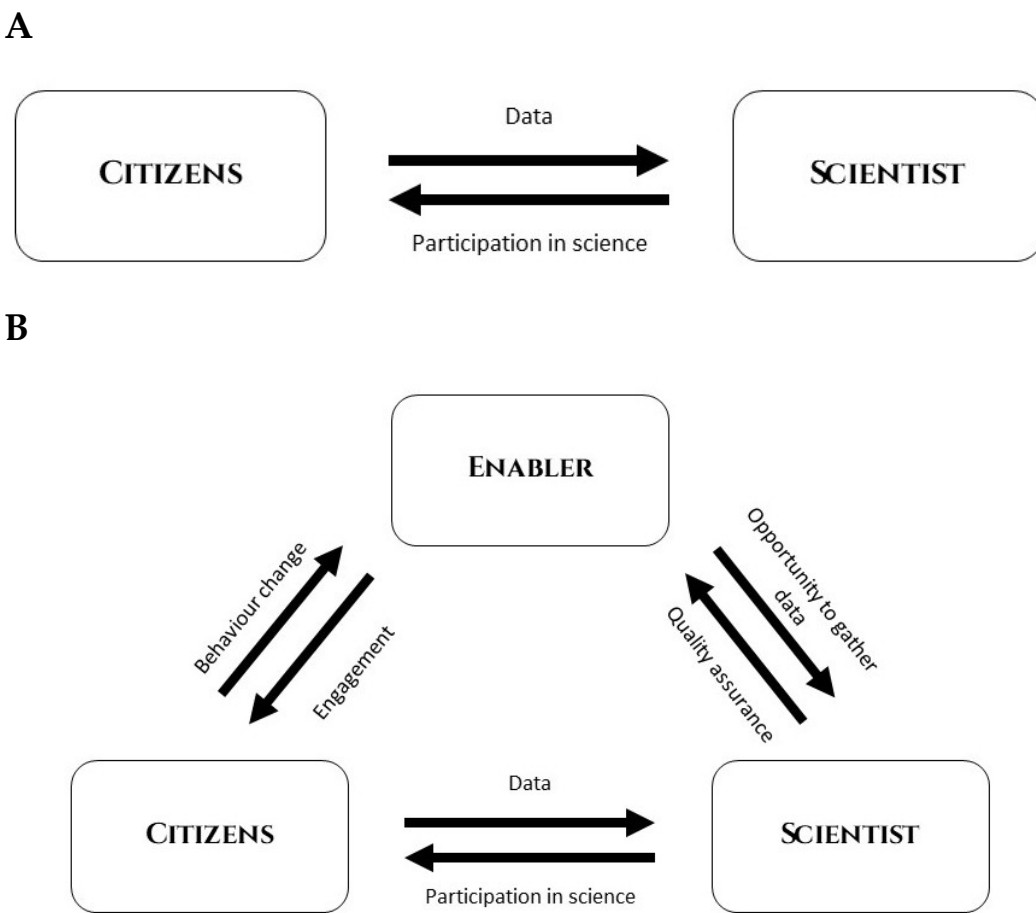

**Figure 1.** Two contrasting representations of Citizen Science: (**A**) The traditional two-party model involving only citizens and scientist. (**B**) An idealised form of the tripartite model of citizen science. Boxes indicate different roles in the project. Arrows indicate what each role is providing to (or receiving from) another, e.g., in (**B**), the enabler provides the citizen with an opportunity to engage whilst the citizen adapts their behaviour in such a way that aligns with the one of the enabler's goals.

Other typologies define projects by their goals and the level of interaction with the physical environment during participation, e.g., Wiggins and Crowston [5] propose a five class categorisation: Action, Conservation, Investigation, Virtual, and Education projects. Citizen science projects have also been categorised according to the form of knowledge acquisition they utilise: sensing, where participants count instances of some event or object of interest; computing, where participants volunteer their personal computers to aid in large scale computing projects; analysing, where participants are asked to identify individuals or objects, or analyse large sets of data collected by scientists or institutions; self-reporting, where participants share their personal records, symptoms, or genomic data, which is gathered for further analysis by scientific institutions, and; making, where participants are asked to be a part of the process of creating new technologies, processes, and knowledge in collaborative 'laboratories' [6]. In Aotearoa New Zealand, an Inventory of Citizen Science classifies projects according to their timeframe: (a) unspecified timeframe, (b) timeframe specified, activities repeated, and (c) one-off events [7].

Haklay, et al. [8] present a thoughtful analysis on the value and ambiguity of definitions in citizen science and the importance of providing associated contexts. In that spirit, we wish to share that all authors of this paper are located in Aotearoa New Zealand and

engaged with citizen science focused on conservation-biology. While the conceptual model we present here may be of value in other domains, our study is focused on conservation problems in citizen science in particular.

Conservation problems and initiatives often feature in citizen science projects. Unlike problems in the 'classical' sciences, conservation problems often include complex social settings and a place-based context where multiple goals and motivations interact via nonlinear trade-offs. These 'wicked' problems and contexts do not have a single universal solution, but rather workable solutions have to be identified for the specific situation [9]. Models of citizen science that are based on traditional scientific problem-solving, are not always effective in these situations. However, methods that create dialogue between scientists and the public, and which encourage scientists to be more reflexive about their activities and actions, can be highly beneficial in these circumstances [10]. Incorporating these latter methods can lead to new ways of conceptualising citizen science projects and understanding how they can contribute to solving wicked conservation problems.

In this context, Mason, et al. [11] provide a particularly relevant framework for considering citizen science through their identification of five themes that contribute to wickedness within conservation-centred citizen science initiatives, as exemplified by a series of diverse conservation conflict case studies and a theoretical framework set out by Game, et al. [9,11,12]. The five themes are: pattern-based evidence and predictive management; distributed decision-making; diverse opinions and creativity; sharing failures, and trade-off based objectives.

It is worth noting that conservation-centred citizen science initiatives can harbour complexity not only within the organisational system, but also within the biological systems of interest. This can also lead to wicked problems. For example, biological invasions have a long-standing association with citizen science, and many projects have been established with the purpose of preventing, monitoring or removing invasive species [13–15]. These projects are encumbered by the wicked nature of biological invasion management arising as a result of a multitude of different factors, including differing stakeholder opinions as to how bad the problem is or what sort of management actions are acceptable, differing stakeholder goals, lack of two-way knowledge dissemination, and complex food web interactions [14,16–18]. To overcome these issues, multiple authors have identified the need for a more collaborative and democratic approach. [14,17,18]. One case in point has been the management of the red lionfish (*Pterois volitans*) introduced into the waters of Belize in 2008. Acceptable solutions were found by involving fishermen, chefs, local people, scientists, conservationists, and representatives of the seafood market [13,19,20].

The case study that was the impetus for this paper, the Great Kererū Count, (described in Section 3) was initially led by two non-governmental organisations, without the involvement of any scientists. The project was initiated with goals of public engagement, education and conservation, with an implicit intent of contributing data to science. After it had been running for three years, two of the authors of this paper (SH and RS) were brought in to analyse the data and assess the scientific value of the project. Initially, the lack of involvement of scientists and some inadequacies of method led them to wonder whether this project was indeed, an "authentic" citizen science project (as they understood it to mean at the time), based on classical models of citizens partnering with scientists and vice-versa. However, the realisation that most of the participants and the wider public clearly considered this to be a citizen science project led the researchers to reconsider their own assumptions of citizen science, explore the various definitions of citizen science in the literature and to reconceptualise how citizen science projects may operate in practice.

This exploration led to the development of a new model for thinking about citizen science, which provides a mechanism to unpack, identify and acknowledge the legitimacy of roles beyond those of the citizen and the scientist, including the differing motivations that might underpin each role. The model can also make explicit any 'wickedness' that is inherent in either the conservation or social context of the initiative. This is in line with Salmon and Roop's framework for designing and interrogating public engagement

initiatives by making explicit inherent power structures in the project, different roles that people play and varying drivers for everyone involved [21].

After providing an overview of this new tripartite model, and its application to three very different conservation case studies, we discuss how it can be applied to wicked problems in conservation-related citizen science more generally.

## 2. The Tripartite Model

While participating in the management and analysis of the Great Kererū Count (GKC) data from 2014 and 2015, we observed first-hand the disconnect between traditional scientific methods and philosophies, and the realities of operating within a multi-party citizen science project [22]. Discussions between parties during a review of the project led to the creation of the tripartite model presented here. The model aims to acknowledge everyone involved in the project, and the different roles that they may occupy: citizen, scientist, and a third, facilitative role, we call the enabler (Figure 1B). It is worth noting that this role may have a better-suited name for a given project. For example, at different iterations of this concept, we referred to it as advocate, educator, enabler and simply 'third party'. Whatever the name, the individual(s) that fill this role are often implicitly involved in citizen science projects as a mechanism to allow scientists to interact with citizens or members of a specific community [6,23–25]. However, the enabler's role in a project often goes beyond this. Recognising the enabler explicitly in the formulation of a project encourages their motivations for being involved to be acknowledged. Taking these motivations into account can allow all people and roles involved to collaborate more effectively. Improving collaboration can subsequently lead to a project running more smoothly, with higher quality data being collected and disseminated to the public, and potentially greater longevity of the project.

Consider, for example, an idealised tripartite model of citizen science, including participants, scientists, and enablers (Figure 1B), each of whom bring goals, skills and opportunities that are critical to the project's success. The scientist provides scientific expertise and the ability to analyse and interpret the data, creating an opportunity for participants to engage in meaningful scientific research and providing quality assurance for the enabler. In our model, the enabler is more than a simple facilitation/communication role because they have their own explicit goals and objectives. For example, this role may be filled by a professional or volunteer educator, advocate, funder or some other third-party. The enabler often brings skills and expertise in facilitation and communication, expertise in public engagement, access to a community or access to funding. These opportunities facilitate the opportunity for participants to engage in the activity and make data-synthesis easier for the scientist. The citizen participants bring skills and opportunities that are central to the other two roles: enabling the project to collect large datasets, which is critical to the scientist, and contributing to behaviour change or growth of a more engaged (and/or active) community, which may be a motivation for the enabler. In its idealised form, none of the roles are marginalised in the service of the others.

A further benefit of the tripartite model is that it provides a mechanism to acknowledge and integrate indigenous or cultural components within citizen science projects. For example, Māori involvement is imperative in many conservation initiatives in Aotearoa New Zealand [26] and application of this model enables their contribution to be formally recognised and considered separately to the scientists and citizens. By including Māori enablers, more relevant indicators and methods may be developed, leading to a range of further benefits for all involved [27]. The value of involving indigenous people in citizen science projects has been explored elsewhere, and includes a subsequent increase in the validity, scope and utility of data gathered [28–30].

The tripartite model is simplistic and presented as a framework for opening dialogue and interrogating a project. It is critical to note that each role in the diagram can be occupied by several people, each with different backgrounds and motivations, and project organization could be unpacked much further. The key insight this model provides is the

ability to make explicit the greater variety of roles and motivations of those participating in a project, beyond the roles of citizen and scientist.

## 3. Case Studies

In order to test the relevance of our model for citizen science initiatives, we applied it to three specific conservation-related case studies with different characteristics. These were (a) the Great Kererū Count, (b) Lion Guardians, and (c) iNaturalist. In each case, we carried out an online interview with a representative of the initiative. The Great Kererū Count interview was conducted initially as a pilot and involved a lead scientist involved in the project (SH, also an author of this paper), the interviewer (RS), and an observer researcher (SR). The Lion Guardians interview involved a director/co-founder of the project, the interviewer (RS) and an observer researcher (SR). The iNaturalist interview involved a co-founder/administrator of the New Zealand branch of iNaturalist, the interviewer (RS) and two observer researchers (SH/SR). All three interviews were conducted online, followed a similar structure (Appendix A) and lasted between 45–90 min.

After asking some introductory questions about the purpose and roles of key individuals in the project, the interviewer attempted to draw a schematic of the different relationships, as guided by the participant, using an iPad Pro (for ease of hand-drawing and real-time virtual screen sharing). After the internal dynamics had been unpacked, examined and documented (taking approximately 45 min), the interviewer shared the proposed tripartite model and discussed with the participant any aspects that resonated (or not) as well as how it could be modified to better represent the project under discussion (Appendix A). While we acknowledge that this approach risks encouraging social desirability bias [31], the main purpose of these interviews was to ground-truth our proposed model and discuss it with colleagues in order to explore its wider relevance, limitations and opportunities for improvement.

We present some key ideas that emerged from those interviews below, drawing attention especially to aspects of inherent wickedness in the projects and resonance or discord with the concept of the tripartite model.

### 3.1. The Great Kererū Count

The Great Kererū Count (GKC) is a citizen science project located in New Zealand, focused on the monitoring of the native pigeon (kererū, *Hemiphaga novaeseelandiae*) distribution and abundance throughout the country (https://www.greatkererucount.nz, accessed on 19 February 2021). Community engagement, education and fostering interaction with nature were also key aspects of the project. The project was created by a non-governmental organisation (NGO) to engage school students with nature, with an implicit aim of gathering data for scientific inference. Scientists were not involved at the outset of the project, but rather were introduced further down the line. Two NGOs and local councils later collaborated and acted as enablers, providing a wider age-range of participants with an opportunity to engage with a scientific project. In doing so, these enablers were hoping for a change in participants' views on nature, increased social media engagement and potentially new members. After three years of running the project, scientists were introduced to analyse the collected data and provide insight to the project, with the hope that the data would provide insight into kererū ecology and distribution. This arrangement is illustrated in Figure 2. Over five thousand participants provided scientists with data (records of occurrence) on a geographical scale that would be otherwise unattainable on usual research budgets. A survey of the participants, conducted at that time, found that 'contributing to science' was one of the key reasons for citizens' participating in the project [22].

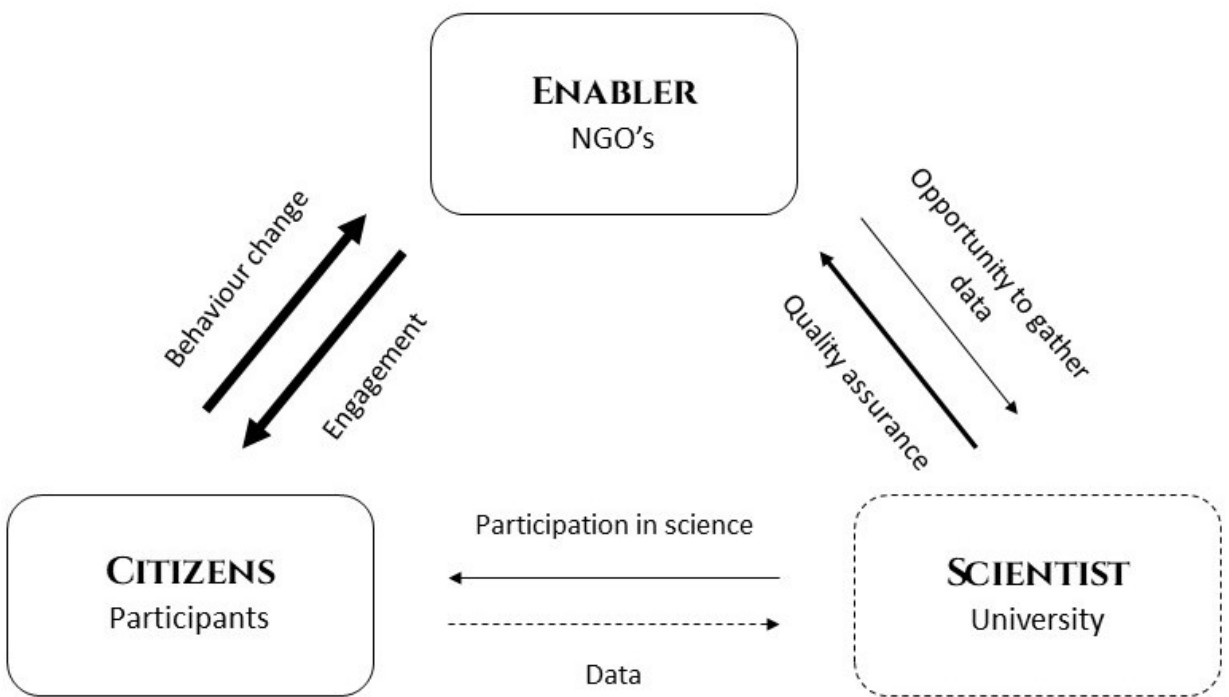

**Figure 2.** The tripartite model of citizen science applied to the Great Kererū Count 2014–2015. Line thickness indicates strength of the provided service. Dashed line indicates slim-to-no service provided. Boxes indicate different roles that individuals might occupy. The project was initiated by two NGOs. Although data was being gathered, scientists were not explicitly part of the initial project formulation.

The initial lack of involvement from a scientist was identified as an initial weakness of the project. Data collection was designed in a way that maximised the ease with which participants could partake, fulfilling the enabler's goals of increasing community participation and social media engagement. This led to the scientific utility of the collected data being unknowingly compromised. At the same time, a survey connected with this particular citizen science initiative [22] identified a strong belief and desire by "a great majority" (over 90%) of participants that the data they collected would be "scientifically valuable". If a scientist had been involved during the design phase, and if the motivations of scientist, citizen and enabler had been explicitly acknowledged, then the end-result could have been more mutually beneficial. That is to say, the scientist could have received more scientifically valuable data, the citizens would have genuinely contributed to the furthering of scientific knowledge, and the advocates could have still achieved their desired increased levels of engagement. These extra steps might well have added to the cost of developing the project. However, the increased utility of data gathered would likely have outweighed the increased costs. Once these different drivers and constraints within the project were recognised, and scientists were engaged to analyse the data, substantial changes were made to the project such that, in subsequent years, the scientific credibility of the Great Kererū Count was enhanced and the motivation to continue and improve was re-affirmed.

*3.2. Lion Guardians*

Lion Guardians is a citizen science project in East Africa that aims to reduce conflict between people and lions by identifying key drivers of conflict and mitigating them (http://lionguardians.org/, accessed on 19 February 2021). To achieve this goal, the program employs local Maasai warriors (teenage boys in the community) as trackers and conflict resolvers. The project was co-created by two scientists, working together with the warriors, committed to reducing conflict between lions and the community in order to reduce the number of lion killings and the number of deaths of both humans and livestock in the region.

During the course of the interview, several roles were identified including scientist-enabler, warrior, funder, board member, project managers and community members. While it would have been relatively simple to present them all in a multi-party model with four or five distinct roles, we used this case study to explore whether there was relevance in trying to force a tripartite framework on the system (or if, for example, we would become an advocate of a multi-partite framework instead).

After some distilling and continued liaising with the interviewee, we identified three key groups: scientists, the warriors and the wider Maasai community. Both the scientists and the warriors play roles as enablers (in different ways) and share the aim of decreasing conflict between lions and the Maasai community in order to reduce the number of livestock killings and protect the Maasai community. In return for this benefit to the Maasai community, the scientists and warriors receive a behavior change from the community, whereby they display an increased tolerance for lions and a decrease in the number of lion killings. In addition, the Maasai community allow the scientists to integrate with the community, which is key to the continuation of the project and gives the community an active and empowered role. The key to the community's behaviour change is the involvement of warriors, who are respected within their communities and create a link with the scientists. In addition to providing data on lion distribution, behaviour and abundance, this link also enhances the scientists' perspective with traditional ecological knowledge (TEK). In return, the scientists provide the warriors with the opportunity to help their community with paid work, an increase in knowledge through schooling in literacy and numeracy and a chance to participate in science (Figure 3).

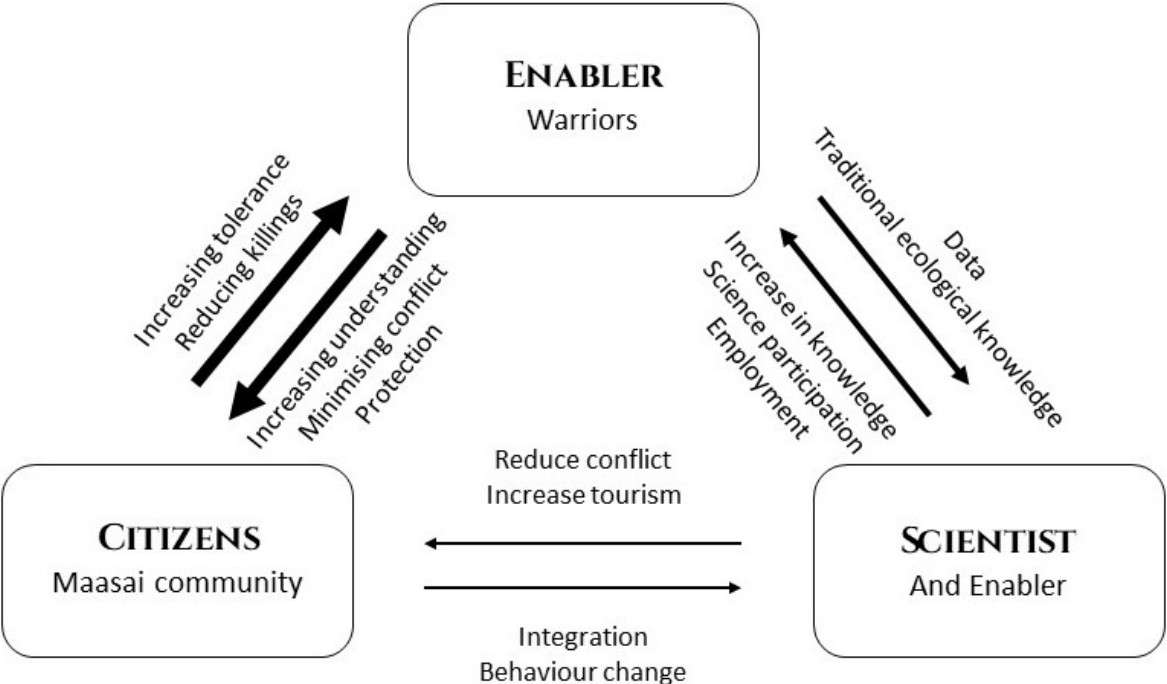

**Figure 3.** The tripartite model of citizen science applied to the Lion Guardians. Line thickness indicates strength of the provided service. Boxes indicate different roles. While the scientist also played a role as an enabler (in the form of an initiator or activist advocating for behavior change), the Maasai warriors were identified as the key enablers that acted as a bridge between the scientist and the wider community.

As with the Great Kererū Count, this system involves various different groups with differing goals. Warriors take part in the project in order to maintain a job and to have a valued role in protecting their community through the use of their ecological skills learned from years of herding and listening to their elders. In addition, warriors relish the opportunity to learn new skills such as reading, writing, and the use of telemetry and GPS units. In contrast, the scientists' primary goals are focused on saving lions. The

community's goal is centered around reducing conflict with lions to protect themselves and their livestock. These differing motivations create inherent wickedness: the scientists' aim to increase lion populations is seemingly incompatible with the community's aims to reduce conflict with lions. However, due to the close collaboration and co-governance between the three parties, and the articulation and acknowledgement of these differing viewpoints, procedures have been put in place, and are continually refined, which accomplish all three goals. This was identified as key to the project's success, and is a key aspect of our model that the interviewee thought was most beneficial to citizen science projects.

This project also differed from the GKC in that some individuals fill several different roles. The lead scientists are not only providing quality assurance, but arealso managers of the project, providing the Maasai warriors the opportunity to gain an education, and facilitating the creation and continuation of the project. This overlap places the project creators in the role of both scientist and enabler. The function of the warriors in this system is even more complex: They are key communicators and mediators between the scientists and the community, making them an important enabler of the project, whilst also being a part of the wider community, filling the role of citizen. However, they are also active data-providers to the scientists, as well as providers of TEK expertise to the scientists.

These overlapping roles may appear to increase the complexity of the project and decrease the utility of the tripartite model. However, whilst any one person may fulfil multiple roles, the contribution of each role to the project remains the same. For example, the enabler still provides a key link between the science and participation by citizens. In this project, the skills required to achieve this are shared between different people: the project creators provide the skills required to educate the citizens, whilst the warriors provide a key cultural link between the scientists' recommendations and uptake of these recommendations by the wider community. The method in which each person contributes to the role may be different but the key aspect of the role, providing the opportunity for the interaction of science and citizenry, remains unchanged.

*3.3. iNaturalist NZ*

iNaturalist is a popular social media platform for naturalists that allows individuals to upload observations of natural phenomena and species occurrences to a collective database. The primary aim of the initiative is to help engage people with nature, with a secondary aim to gather data (https://inaturalist.nz/, accessed on 5 May 2021). Within the framework of the record gathering and databasing, a plethora of projects with more specific aims can be created. Scientists are involved with iNaturalist in a quality assurance role, where they provide clarification to citizens on species' identities and advise on how projects can be constructed to maximise the usefulness of the data. Experienced amateur naturalists also frequently fulfill these roles. The advocacy role is filled by project managers who administrate individual programs within iNaturalist. Participants supply observations on species and phenomena of interest.

Due to the broad scope and scalability of iNaturalist, our interview identified that multiple roles are often filled simultaneously by a single individual, making them both enabler and scientist, enabler and participant, participant and scientist, or all three. This led to a discussion about how the model could be adapted to account for individuals fulfilling multiple roles simultaneously. Modification of the model into a Venn-diagram allowed us to visualize this overlap and provided a more realistic representation of iNaturalist's structure (Figure 4).

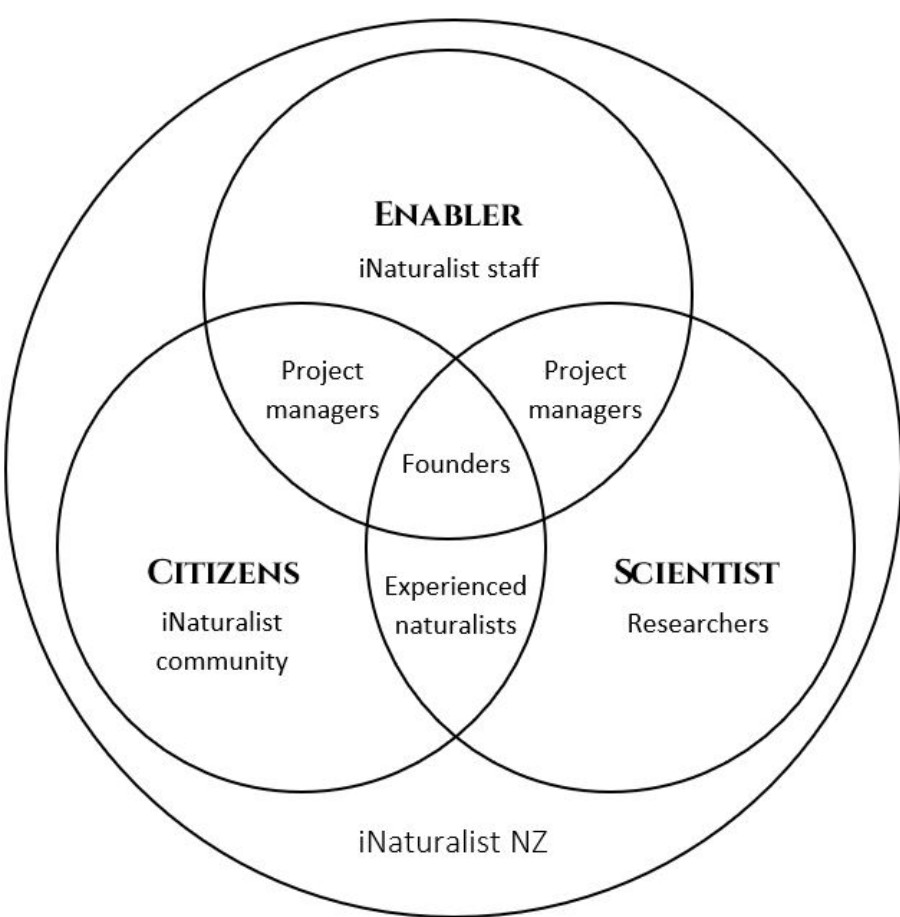

**Figure 4.** The tripartite model of citizen science applied to iNaturalist. Circles represent roles. Due to the nature of iNaturalist, individuals often occupied more than one role. Adapting our figure into a Venn-diagram allows us to visualise this overlap. Services provided by each role remain similar to Figure 1B. In this case enablers provide project management and technical services to the iNaturalist platform. We have omitted the inclusion of arrows in this case, in order to focus on the key insightof this case study, which was an improved depiction and acknowledgement of overlapping roles.

The steering group and developers of the platform aim to maintain a large number of users, and user accessibility through the collection of standardised, basic data fields, whilst scientists frequently design sub-projects within iNaturalist that require collection of a greater number of nuanced data fields. These two aims can create tension as keeping people engaged requires a simple interface with minimal rules for data collection, whilst high quality data collection requires systematic sampling techniques and increased scientific rigour. Due to this, many projects within iNaturalist do not reach their secondary goal of providing useful scientific insights.

Collaboration between scientists and participants or project managers at an early stage could encourage projects to be created in a way that can achieve the goals of both perspectives: high quality data for the scientist and engagement for the project manager and citizens [32]. Successful projects that achieve these aims are often co-designed by citizens and scientists, and bubble up from citizens' interest in a subject [32]. More successful projects might be co-created through the integration of our model at the project's outset. For example, bio-blitz projects are often organised by community members, who also act as project managers, and aim to collect a large amount of data in a small area, in a small amount of time (e.g., school grounds, recreational park, or reserve) [33]. Many of these projects are conducted in an indiscriminate manner, leading to data that is not highly valued or used by scientists (despite participants believing the data will be useful). Incorporating a scientist in the early stages of the project's conception could allow project

managers to understand how the project needs to be conducted to acquire useful data. Project managers can then find a balance between a project that is interesting and engaging, but also fulfils its secondary aim of gathering meaningful data.

## 4. Discussion

The case study interviews gave us confidence that a tripartite model could be a helpful lens through which to consider or interrogate a given citizen science initiative. In addition, the flexibility of the model was found to be beneficial as it allowed us to adapt it to represent each project more accurately and specifically. Both the Lion Guardians and iNaturalist projects highlighted involved individuals who occupied two or more roles. This helped us to expand our concept of the model, exploring alternative visualisations, such as the use of a venn-diagram, in order to represent these overlaps more clearly.

Using the model to describe a project provided a framework for unpacking and making explicit inherent complexities and wickedness, both within the organisational structure and the social and biological systems under examination. These aspects are explored in more detail below, prior to a consideration of the limitations of this study and future opportunities for extending this research.

### 4.1. Organisational Complexity/Wickedness

Mason, et al. [11] identified five themes that contribute to wickedness within conservation-centred citizen science initiatives, four of which can be usefully explored by our tripartite model.

(i)  Distributed decision making: One of the recurrent issues within citizen science projects, as identified in both the Great Kererū Count and Lion Guardians, is the presence of conflicting goals held by scientists, citizens, and the implicit enabler [9,11,23,25,34]. We can raise awareness of these motivations, and how they may conflict with each other, by explicitly acknowledging the multiplicity of roles and relevant people within and throughout the citizen science process. Acknowledging and utilising the enabler role as a link between scientists and citizens increases the movement of information in all directions. Increasing the movement of information allows those involved to collaborate more effectively, creating a democratic and distributed decision-making process [35,36].

(ii)  Diverse opinions and creativity: The sometimes linear, traditional thinking of working scientists and their belief that citizens' ability to contribute is limited to accessing larger datasets [23] can stymy the potential of citizen science projects to make impactful change in the world. Increasing the creative potential of a project can be achieved by including everyone with a vested interest in the project in the decision-making, project design and the scientific process [37]. Furthermore, by ensuring the project is as democratic as possible, no single person or role exerts a dominant influence, quelling other diverse and creative contributions [38]. This is particularly important when working with indigenous cultures as respecting and integrating their ideas and worldviews enables a wider perspective and a wider set of values to be included in the process. Incorporating this perspective into a project allows a greater range of information to be gathered and increases the efficacy of human-wildlife conflict management [28–30,37].

(iii)  Sharing failures: A ubiquitous feature across many citizen science projects is the opacity surrounding real or perceived failures [11,12,34]. Hiding failures creates an environment where best-case solutions are unattainable. Formative evaluation of a citizen science initiative through the lens of the the tripartite model provides a mechanism to encourage transparency and discuss weaknesses in the system, with a goal of trying to better deliver on the goals of all individuals involved. This will hopefully increase the ease with which best-case solutions can be found by understanding failures, or misaligned goals, from the perspective of all involved. Our experience with the GKC is an excellent illustration of this; the identification of the

misaligned motivations that different roles held led to tangible changes that enabled future iterations to better deliver on the goals of the scientist and citizen without compromising the goals of the enabler.

(iv) Trade-off-based objectives: Each of our case studies had multiple objectives, dependent on the differing aims of those involved. In some cases, the objective of one role was fulfilled to the detriment of another role's objective. The most pertinent example of this is the simplification of data-gathering in the GKC. This helped achieve the enabler's objective of increasing citizen uptake to the detriment of the scientist's objective of gathering high quality data. Similar tensions arise in many iNaturalist projects. To ensure no individual's objective is fulfilled to the detriment of another's Mason, et al. [11] suggest acknowledging differing aims and objectives throughout the project, from conception through conclusion. The tripartite model encourages everyone involved to make explicit their aims and objectives. This makes it easier to acknowledge the differences and conflicts between objectives, allowing the best case trade-offs to be identified.

The key benefit of our tripartite model is its ability to increase democratisation and balance in a project by encouraging knowledge dissemination and dialogue between everyone with an invested interest. The acknowledgement of those in enabler roles creates a more holistic and realistic perspective of the whole project, reduces power differences between individuals, and can aid the remediation of differing stake-holder values; the feature identified by Mason, et al. [11] as a key contributor to conservation conflict and wicked problems in conservation projects [11].

### 4.2. Wickedness Resulting from Complexity in Socio-Biological Systems

Many conservation-centred citizen science projects harbour complexity, not only within their organisational structure, but also within the biological systems, leading to socio-biological challenges that are inherently 'wicked'. These issues can arise out of differing stakeholder opinions, differing stakeholder goals, lack of knowledge-sharing and complex food web interactions [14,17,18]. While, the complexity of food web interactions cannot be adequately resolved through our model, the former three factors can, in part, be addressed by the tripartite approach. Increasing collaboration and co-governance (recommended in these papers) can be achieved by applying our tripartite model and explicitly acknowledging the non-traditional role(s) within citizen science projects. This allows all individuals involved in the project to introduce and acknowledge their aims, and motivations for undertaking the project from the outset. In the case of biological invasions, for example, the enabler may be a government ministry or a particular industry that has their own reasons for wanting surveillance of a perceived pest species to be a priority.

A tripartite approach has a two-fold impact on the consideration and design of invasive-species citizen science projects in particular. Firstly, the inclusion and explicit acknowledgement of an enabler in these projects typically facilitates the dissemination of knowledge between citizens and scientists. Most invasive species citizen science projects involve some knowledge dissemination, either from citizen to scientist [30], or from scientist to citizen [39]. A dedicated specialist in the realm of communication between scientist and citizen can increase and enhance knowledge sharing, thereby creating a more dynamic environment where everyone who is involved in the project is acknowledged and able to provide input. Expanding citizens' knowledge about the project allows them to take an active role in the decision-making process from design through the implementation of decisions. This creates more robust projects, and enables outcomes that benefit the scientific community, decision makers, citizen scientists and the wider community [17].

Secondly, a tripartite (or multi-party) model, which explicitly names the flows of benefits between parties, acknowledges that different stakeholders may have different values, aims and motivations regarding the (invasive) species of interest [40,41]. What is damaging for one group of people may be a valued resource for another. For example, the invasive red lionfish in Belize led to significant reductions in native fish populations as

well as damage to coral reefs, resulting in a decrease in the profitability and subsistence potential of fishing for the local people [20]. Through the discussions with many different groups (fishermen, chefs, local people, scientists, conservationists and representatives of the seafood market), it was agreed that the best long-term solution was to create a market for red lionfish [19]. This collaborative approach took into consideration the aims of all parties including profitability for the seafood market and fishermen, as well as a reduction in lionfish numbers and an increase in native fisheries, which was the motivation for scientists and conservationists [13,19]. The additional measures were subsequently introduced to conserve marine reserves where fishing was not allowed, including using citizen scientist SCUBA divers to monitor native fish populations and cull lionfish [13].

*4.3. Limitations and Future Directions*

In this paper, we have proposed a new conceptual model for unpacking complexity and wickedness within conservation-centred citizen science initiatives. This grew out of our experiences within a citizen science initiative and was further 'sense-checked' against two quite different conservation-centred citizen science projects. This was not therefore a methodologically robust test of the model, but rather an exploratory process of making-meaning of our own and our colleagues' experiences. We anticipate that the ideas in this paper may resonate with citizen science researchers, practitioners, enablers and participants, and we hope that the model presents a new framework that is helpful for unpacking, planning and clarifying inherent dynamics, expectations and goals within a given project. A more robust analysis of this model would, for example, require interviews from many more projects, and include individuals who play different roles (e.g., enabler, scientist, citizen or a combination of these) within a given project.

Future work should focus not only on the data that interrogates the concept, but also the design of the model itself. For example, the information shared within the diagrams, the relative positions of the different roles, clarification of the more nuanced role of individuals, and the use of different design effects to signify the different equilibria that exist within a project.

In this paper, we primarily argue for the need to acknowledge roles that occur in citizen science initiatives beyond the citizen and the scientist, using case studies from conservation biology. Future work could therefore also explore the value of expanding this tripartite model to a multi-party one. In addition, we are interested in observing the potential value of this approach when applied to citizen science projects that address social issues and concerns beyond conservation projects, for example questions related to poverty, health and housing.

## 5. Conclusions

Citizen science projects frequently suffer from inherent wickedness due to conflicting goals and motivations and/or lack of support for all key roles. Our tripartite model strives to address this wickedness by providing a framework to make explicit, and acknowledge, the contributions and motivations of the multiple roles needed for a successful project. In particular, we highlight the contribution of an often over-looked and little discussed role, that of the 'enabler' (who may also be recognised as an educator, communicator, change advocate etc.). Explicit dialogue between all invested individuals, both at the outset of a project and as a component of ongoing reflection and improvement, brings to the fore any incongruence between their goals and motivations, as well as inherent power structures. Solutions that satisfy everyone involved can then be discovered, allowing the project to run more smoothly, encourage wider and deeper citizen engagement, and collect scientifically valued data that can be used to inform management of a wide range of conservation issues.

**Author Contributions:** Conceptualisation, R.A.S., S.H. and M.T.E.; methodology, R.A.S., S.H., M.T.E. and S.R.; investigation, R.A.S., S.H. and S.R.; writing—original draft preparation, R.A.S., S.H., M.T.E. and S.R.; writing—review and editing, R.A.S., S.H. and S.R.; visualisation, S.R.; supervision, R.A.S.

and S.H.; funding acquisition, R.A.S., M.T.E. and S.H. All authors have read and agreed to the published version of the manuscript.

**Funding:** Victoria University of Wellington and Wellington City Council. S.H. acknowledges additional support from Ministry of Business, Innovation and Employment grant UOWX1601 422 (People, Cities and Nature).

**Institutional Review Board Statement:** Not Applicable.

**Informed Consent Statement:** Informed consent was obtained from all subjects involved in the study.

**Data Availability Statement:** The data presented in this study are available on request from the corresponding author. The data are not publicly available due to privacy restrictions.

**Acknowledgments:** We acknowledge Stephanie Dolreny and Jon Sullivan for sharing their insights into the Lion Guardians and iNaturalist case studies, respectively as well as early interviews with organisers of the Great Kererū Count 2014. Summer Scholars and Research Assistants: Anni Brumby and Rebecca Calder-Flynn collated data and contributed to the development of our ideas. We also thank three anonymous reviewers for their suggestions. Finally, we thank the scientists, enablers and most of all the many thousands of citizens who have participated in any of the projects featured in this article.

**Conflicts of Interest:** The authors declare no conflict of interest.

### Appendix A. Tripartite Framework—Interview Questions for Case Studies

1. Introductory statement around ethics, use of data, recording, feedback and background to people participating in the interview team, including their connection to this research.
2. Can you please tell us a bit about [case study]?
3. What would you say is its main purpose or goal?
4. How did it come about?
5. One of the things we've been thinking about with this research are elements of complexity and messiness in Citizen Science, as well as "wickedness", as in wicked problems, which are problems that are impossible to solve. This could anything from complexity in the scientific or ecological system being studied, to complexity due to the interactions and power structures between individuals and organisations who are involved. Can you talk a bit about different elements of complexity or messiness within [case study]?
   - Within the science or research area that you are studying
   - Within the organisation (for example, as a result of having multiple players)
6. I'm going to try and document this visually on a shared screen while you talk, but you might also end up just scribbling things on a piece of paper and holding it up to the camera! Can you describe the key actors or players in the [case study]? What roles do they each play? In what ways do they interact with each other, or serve and support each other? Then try to draw it and talk it through.
7. [Only after identifying key actors, dynamics and relationships.] This research built out of an experience that [SH and RAS] had with a citizen science project where we realised that all the actors involved had very different drivers and rewards, and also played critical roles (and sometimes also wore several hats). We came up with the idea of a tripartite framework, which had citizen, scientist, and also advocate or enabler. You can see in this diagram how information flows and how they all support each other in an idealised system. Does this have any resonace for you? Are there roles that you could map onto this, or any additional roles you would add to this?
   a. Does anyone have multiple roles, and how does that influence the internal dynamics?
   b. How might this diagram be adapted to better represent [case study], if there is any relevance at all?
8. Was this conversation at all insightful or useful, and if so, in what way?

9. Thanks! I might follow up again in a few weeks, or feel free to ping me if you find the conversation triggers any new reflections.

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
