# Peer review of "Citizens, Scientists, and Enablers: A Tripartite Model for Citizen Science Projects"

_diversity, doi:10.3390/d13070309_

Round 1

Reviewer 1 Report

First of all, I would like to emphasise that the approach the authors take by introducing an additional role to the reflection of citizen science projects brings an interesting focus into debates in citizen science.

The manuscript is nicely written and I could follow the argument through most parts of the text.

I do encourage the authors to make the effort and revise the manuscript as it would add nicely to the discussions about CS.

My recommendations for improvement of the manuscript address different issues:

  1. Concept of tripartite model

There seems to be an inherent difficutly resulting from the fact that citizens, scientists and enablers are put to the same level to focus on different roles in citizen science projects. However, 'citizens' and 'scientists' as terms do not describe the roles people take in a CS project - whereas 'enabler' clearly is a term for a specific role or function of a person. Through the case study of Lion Guardians, the authors show that ppl in a CS project can have overlapping roles.

I would recommend some further reflections on the issue of roles, yet, I can see that this is a wicked problem that needs some more disentangling of current terminologies and terms for ppl involved in a CS project.

Lines 106-119 needs more in-depth description of roles in CS projects. The authors jump too quickly on the role of enabler and describe the consequences (lines 114-119). It would be very interesting to see different roles that could be fulfilled in CS projects, no matter who takes them.

Line 126-127 - a role cannot self-define, please adapt the phrasing.

Line 133 + 144 and graph B 'contribute to behaviour change' - How? Whose behaviour? Please be more explicit.

Line 144 - odd phrasing: 'the citizen provides the enabler with some form of ...' What does that mean?

Lines 146-152 - needs referencing and evidence.

2. Case studies - methodological problems

The authors interviewed one representative of each case study which leads to anecdotal evidence and as such cannot serve for drawing wider conclusions or be transferred to other studies.

The argument would have been much more convincing if the authors had interviewed one representative for each role (per case study) that they refer to in their tripartite model. I would strongly recommend the authors do so to increase the evidence and argument they build and advocate for: to increase the dialogue and make space for actors' motivations and roles in a CS project.

The authors state that in the course of the interview, they showed the tripartitite model to interviewees and asked them how it resonated (or not) with participants. Here, we clearly face the problem of social
desirability bias which as a phenomenon in interviews has been widely describes and discussed in scientific discourse.

Overall, the manuscript lacks a critical reflection of methods which needs to be added in the discussion at the latest.

Case study 1 advocates for scientists being involved in the project for more robust data. However, it is not clear to me how this adds to the tripartite model.

Lines 207-208 - please explain how/in what way the end result would have been more beneficial.

Case study 2 - Here, the authors have the chance to disentangle the different roles even one actor group can take, i.e. the warriors. In addition, it is not quite clear which role the community takes. From the descrition I read that the community takes the role of onlookers to the project however, they might have a strong saxing in the end results. The warriors take the role of enablers, active data providers, communicators/mediators etc. This is such a nice example and could be elaboratted more in line with the main arguemtn of the manuscript.

Case study 3 - the graph is too symplisitc and should deliver more information. Again, see above, actors are named but not roles. Please be much more specific.

3. Discussion

The discussion should reflect the findings against exisiting literature which is done against the framework of Mason et al which could have been introduced earlier in the manuscript.

4.1 Organisational wickedness

The framework by Mason should be introduced earlier on in the manuscript and be referred to later on. Moreover, as a reader I am wondering what the two other themes are and why they do not apply to your model?

(iii) should be elaborated more. It is not yet convincing as it stands. Where and how does your argument provide information about sharing failures? If you connected this to an argument earlier on in the manuscript where you might speak about different roles and which acotrs in CS projects could take, then it would fit better.

4.2 Biological complexity

The first paragrpah of this section (376-394) is not clear to me. What do you mean by 'biological systems of interest'? Why do you raise the issue of invasive species, food web connections etc at this point? To me, this first paragraph still talks about governance/organisational complexity which is fair but the connection to biology etc confuses me as a reader.

The discussion should not bring in new case studies (e.g. lines 400-411) but rather discuss limitations of the current analysis, including methodology and transferability of results.

5. Introduction

Line 40 - please avoid terminology like 'utilize' citizens - wording matters, see e.g. Eitzel et al. 2017

Lines 87-93 - interesting observation. The following references might be of interest to you and could be reflected in the introduction:

Haklay, M., et al. (2021). What Is Citizen Science? The Challenges of Definition. The Science of Citizen Science. K. Vohland, A. Land-Zandstra, L. Ceccaroni et al. Cham, Springer: 13-33.

Haklay, M., et al. (2020). "ECSA's Characteristics of Citizen Science."
   DOI: 10.5281/zenodo.3758668

Author Response

Point 1: Concept of tripartite model

There seems to be an inherent difficutly resulting from the fact that citizens, scientists and enablers are put to the same level to focus on different roles in citizen science projects. However, 'citizens' and 'scientists' as terms do not describe the roles people take in a CS project - whereas 'enabler' clearly is a term for a specific role or function of a person. Through the case study of Lion Guardians, the authors show that ppl in a CS project can have overlapping roles.

I would recommend some further reflections on the issue of roles, yet, I can see that this is a wicked problem that needs some more disentangling of current terminologies and terms for ppl involved in a CS project.

Lines 106-119 needs more in-depth description of roles in CS projects. The authors jump too quickly on the role of enabler and describe the consequences (lines 114-119). It would be very interesting to see different roles that could be fulfilled in CS projects, no matter who takes them.

Line 126-127 - a role cannot self-define, please adapt the phrasing.

Line 133 + 144 and graph B 'contribute to behaviour change' - How? Whose behaviour? Please be more explicit.

Line 144 - odd phrasing: 'the citizen provides the enabler with some form of ...' What does that mean?

Lines 146-152 - needs referencing and evidence.

Response 1:

We thank Reviewer 1 for these insights and acknowledge that further and more concise clarification was required between “actors” and “roles”, as well as the complexity and inherent wickedness between these. We have clarified our language and use of these terms throughout the manuscript and included an additional paragraph defining how we are using these terms. Further, building on Voronov and Weber (2020), we have removed all reference to actors throughout the manuscript, preferring instead to acknowledge the individual people and the various roles they occupy, and expertises they hold, both within and without the citizen science project under discussion.

This includes clarification of terminology and expanding meaning in lines 106-119, 126-127, 133 and 144 (see track changes for details, although these original line numbers have changed as a result of the edits).

We have included two further references in [the original] lines 146 – 152 to support and strengthen our argument.

Point 2. Case studies - methodological problems

The authors interviewed one representative of each case study which leads to anecdotal evidence and as such cannot serve for drawing wider conclusions or be transferred to other studies.

The argument would have been much more convincing if the authors had interviewed one representative for each role (per case study) that they refer to in their tripartite model. I would strongly recommend the authors do so to increase the evidence and argument they build and advocate for: to increase the dialogue and make space for actors' motivations and roles in a CS project.

The authors state that in the course of the interview, they showed the tripartitite model to interviewees and asked them how it resonated (or not) with participants. Here, we clearly face the problem of social desirability bias which as a phenomenon in interviews has been widely describes and discussed in scientific discourse.

Overall, the manuscript lacks a critical reflection of methods which needs to be added in the discussion at the latest.

Case study 1 advocates for scientists being involved in the project for more robust data. However, it is not clear to me how this adds to the tripartite model.

Lines 207-208 - please explain how/in what way the end result would have been more beneficial.

Case study 2 - Here, the authors have the chance to disentangle the different roles even one actor group can take, i.e. the warriors. In addition, it is not quite clear which role the community takes. From the descrition I read that the community takes the role of onlookers to the project however, they might have a strong saxing in the end results. The warriors take the role of enablers, active data providers, communicators/mediators etc. This is such a nice example and could be elaboratted more in line with the main arguemtn of the manuscript.

Case study 3 - the graph is too symplisitc and should deliver more information. Again, see above, actors are named but not roles. Please be much more specific.

Response 2:

We thank Reviewer 1 for this useful feedback, and wholeheartedly agree that a more robust interrogation of this proposed model is necessary, including interviews from a wider range of case studies, and representatives from each of the roles involved. We have included an additional paragraph in our discussion recommending future directions for this research [now at line 510 – 533], and thank the reviewer for this suggestion. We have also strengthened our language to clarify that this manuscript is primarily a “think-piece”, with a few case studies provided as illustrations, and in no way an in-depth analysis or investigation of the model, which would require a far more robust methodological approach. We would be delighted if readers of this article found sufficient value in our proposed model that they chose to interrogate it further in this way.

We also fully recognise the problem of social desirability in our methodology, and have made this acknowledgement more explicit in the manuscript [lines 220 – 223]. Again, we reiterate that the purpose of the discussions with the interviewees was not to test the model, but rather to explore if it had any resonances with their lived experiences.

Building on these examples, we have further emphasized that this paper is not trying to be “methodologically robust” in terms of its analysis and interrogation of the model (which would warrant a whole new research project), but rather is providing a new conceptual framework for considering citizen science initiatives.

With regard to the comments on specific lines,

  • [Initial lines 207 – 208] We have clarified the context in case study 1 and made explicit ways that the end result would have been more beneficial [now at lines 247 – 265].

  • We have elaborated on the dynamics in case study 2, as suggested [now at 317 – 336]

  • We have added more information into the diagram associated with case study 3, including more information on roles as well as actors, as suggested [Fig 4].

Point 3. Discussion: Organisational wickedness

The discussion should reflect the findings against exisiting literature which is done against the framework of Mason et al which could have been introduced earlier in the manuscript.

The framework by Mason should be introduced earlier on in the manuscript and be referred to later on. Moreover, as a reader I am wondering what the two other themes are and why they do not apply to your model?

(iii) should be elaborated more. It is not yet convincing as it stands. Where and how does your argument provide information about sharing failures? If you connected this to an argument earlier on in the manuscript where you might speak about different roles and which acotrs in CS projects could take, then it would fit better.

Response 3:

We have introduced the Mason literature earlier in the manuscript as suggested [now at lines 90 – 96], mentioned all of the themes and explored the relevance of each to our model [ 408- 466, as before, but expanded]. All literature explored in the discussion has therefore now been introduced and situated in the introduction.

We have also elaborated more on (iii) sharing failures, by connecting with examples in the case studies [443 – 447].

Point 4. Discussion - Biological complexity

The first paragrpah of this section (376-394) is not clear to me. What do you mean by 'biological systems of interest'? Why do you raise the issue of invasive species, food web connections etc at this point? To me, this first paragraph still talks about governance/organisational complexity which is fair but the connection to biology etc confuses me as a reader.

The discussion should not bring in new case studies (e.g. lines 400-411) but rather discuss limitations of the current analysis, including methodology and transferability of results.

Response 4:

We thank the reviewer for raising these points, as they identified some assumptions that we had not made explicit in this paper. To address this, we have moved the introduction of this literature to the introduction of the paper [ now at lines 97 – 111] and provided a stronger explanation that we are focussing our study on conservation-centred citizen science [lines 72 – 77]. This is both due to our own disciplinary experience, but also due to the nature of the Special Issue that we submitted this article to, which is focussed on wicked problems in conservation citizen science. We have also adapted the discussion component to reflect this and focus on the opportunities and limitations of this model, rather than introducing new literature at that stage.

We have also expanded on the limitations of our methodology and transferability of the results [lines 510 – 533].

Point 5. Introduction

Line 40 - please avoid terminology like 'utilize' citizens - wording matters, see e.g. Eitzel et al. 2017

Lines 87-93 - interesting observation. The following references might be of interest to you and could be reflected in the introduction:

Haklay, M., et al. (2021). What Is Citizen Science? The Challenges of Definition. The Science of Citizen Science. K. Vohland, A. Land-Zandstra, L. Ceccaroni et al. Cham, Springer: 13-33.

Haklay, M., et al. (2020). "ECSA's Characteristics of Citizen Science."
   DOI: 10.5281/zenodo.3758668

Response 5

We have changed our use of language in line 40 [still at line 40] and expanded on the references [ now at lines 72 – 77]. Many thanks for these suggestions, which were very helpful. We thank this reviewer in particular to directing us to Hacklay (2021), which provided a very useful framing for our literature review and introduction, and which we have now cited in our paper.

Reviewer 2 Report

The paper present a very interesting approach to Citizen Science.  It is valuable to investigate the role of enablers in Citizen Science, a role that has been neglected for too long, and more generally, the role of third parties who are not stricly speaking "the scientists" or "the citizens". The case studies are interesting and well-presented.

I propose few minor suggestions to improve the paper :

  • add in supplementary information examples of visual documentation obtained during the interviews, as drawn by the persons interviewed
  • discuss the validity of a tri-partite model versus a multi-partie one ; why 3 elements form a good model (whether they play distinct of overlapping roles) and why a more multidimensionnal approach is not preferablee
  • be clearer in the introduction and conclusion, and maybe in the discussion, that although your case studies are focused on ecological / conservation issues, your proposal is framed in more general terms, that I beleive you want to propose as useful also for citizen science programs not ecology / biology related. E.g questions related to poverty, health, housing, etc...
  • I propose that the size of the boxes for scientists / enablers / citizens could be used to represent different equilibria in the projects, for example in terms of number of people involved
  • I suggest that you should rotate slightly the diagrams to avoid having the "enablers" on top and central.

Author Response

Point 1: add in supplementary information examples of visual documentation obtained during the interviews, as drawn by the persons interviewed

Response 1:

This is an excellent idea, and one that we would utilise in future. However, at the time of the interview we adopted a “full confidential disclosure” approach, which included naming individuals in order to better interrogate processes within the case study. These sketches and pictures therefore include confidential information that we are not at liberty to share and would require a more robust ethics process that cannot be carried out retroactively.

Point 2: discuss the validity of a tri-partite model versus a multi-partie one ; why 3 elements form a good model (whether they play distinct of overlapping roles) and why a more multidimensionnal approach is not preferable’

Response 2:

This is an excellent recommendation, which we have incorporated into the new section on Future Directions [lines 510 – 533].

Point 3: be clearer in the introduction and conclusion, and maybe in the discussion, that although your case studies are focused on ecological / conservation issues, your proposal is framed in more general terms, that I beleive you want to propose as useful also for citizen science programs not ecology / biology related. E.g questions related to poverty, health, housing, etc...

Response 3:

We have clarified the boundaries of this study and provided a stronger explanation that we are focussing our study on conservation-centred citizen science [lines 72 – 77]. This is both due to our own disciplinary experience, but also due to the nature of the Special Issue that we submitted this article to, which is focussed on wicked problems in conservation citizen science.

We have also added a new section [Future Directions, lines 510 – 533], in which we acknowledge the potential wider application for the model and opportunity for future research.  

Point 4: I propose that the size of the boxes for scientists / enablers / citizens could be used to represent different equilibria in the projects, for example in terms of number of people involved

Response 4:

This is a valid suggestion, and one that we have included as a possible direction that further exploration of this model could take. However, in order not to overcomplicate the diagrams we have elected to keep the boxes a constant size so that we focus on the interactions between the roles. We have also taken extra efforts to make clear that this is primarily a “think-piece” proposing a new conceptual model for citizen science, and with scope for significant further finessing and refining in the event that the model resonates with readers, CS practitioners and fellow CS researchers [lines 511 – 520].

Point 5: I suggest that you should rotate slightly the diagrams to avoid having the "enablers" on top and central.

Response 5:

Thank you for this suggestion, which we understand. We explored several alternate graphical options in response to this suggestion, but struggled to create one in which any role did not appear to dominate. We chose in the end to keep enablers at the top, primarily to illustrate the visual progression from the original two-party model in Figure 1. We have, however, adapted all of the diagrams now so that citizens appear on the left, and scientists on the right, which we feel strengthens the position of the citizen and reduces the strength of the position of the scientist. In addition, we have included a recommendation that future work would pay closer attention into the design components of the model so that the better represent the concepts we are attempting to communicate [lines 526 – 530].

Reviewer 3 Report

In this paper, the authors propose a theoretical framework to analyse citizen science projects: a tripartite model composed of citizens, scientists, and an important third actor, which they call the “enabler”. The authors consider that this third element incorporates additional expertise and skillset often present in citizen science, but frequently overlooked in associated literature.

The model is evaluated in three case studies to explore up to which point the success and sustainability of a citizen science project require all actors to be acknowledged and interacting appropriately.

In my view, the results demonstrate the utility of this new formalization, which I think will be used extensively in future citizen science-related literature. For this reason, I recommend the publication of this contribution

Author Response

Point 1. In this paper, the authors propose a theoretical framework to analyse citizen science projects: a tripartite model composed of citizens, scientists, and an important third actor, which they call the “enabler”. The authors consider that this third element incorporates additional expertise and skillset often present in citizen science, but frequently overlooked in associated literature.

The model is evaluated in three case studies to explore up to which point the success and sustainability of a citizen science project require all actors to be acknowledged and interacting appropriately.

In my view, the results demonstrate the utility of this new formalization, which I think will be used extensively in future citizen science-related literature. For this reason, I recommend the publication of this contribution.

Response 1.

We would like to thank Reviewer 3 for this supportive review, which has further encouraged us in our belief that we are suggesting a useful new conceptual model for citizen science.